# Evaluation of performance of two SARS-CoV-2 Rapid IgM-IgG combined antibody tests on capillary whole blood samples from the fingertip

**Thierry Prazuck[1]\*, Mathilda Colin[1], Susanna Giachè[1], Camélia Gubavu[1], Aymeric Seve[1], Vincent Rzepecki[1], Marie Chevereau-Choquet[1], Catherine Kiani[1], Victor Rodot[1], Elsa Lionnet[1], Laura Courtellemont[2], Jérôme Guinard[2], Gilles Pialoux[3], Laurent Hocqueloux[1]**

**1** Department of Infectious and Tropical Diseases, CHR Orléans, Orléans, France, **2** Department of Virology, CHR Orleans, Orléans, France, **3** Department of Infectious Diseases, Hôpital Tenon, Assistance Publique des Hôpitaux de Paris, Paris, France

\* thierry.prazuck@chr-orleans.fr

**Data Availability Statement:** All relevant data are within the manuscript and its Supporting Information files.

## Abstract

### Background

The SARS-CoV-2 (Severe Acute Respiratory Syndrome CoronaVirus 2) is responsible for the infectious respiratory disease called COVID-19 (COronaVIrus Disease 2019). In response to the growing COVID-19 pandemic, point-of-care (POC) tests have been developed to detect specific antibodies, IgG and IgM, to SARS-CoV-2 virus in human whole blood. We conducted a prospective observational study to evaluate the performance of two POC tests, COVID-PRESTO® and COVID-DUO®, compared to the gold standard, RT-PCR (real-time reverse transcriptase polymerase chain reaction).

### Methods

RT-PCR testing of SARS-Cov-2 was performed from nasopharyngeal swab specimens collected in adult patients visiting the infectious disease department at the hospital (Orléans, France). Capillary whole blood (CWB) samples from the fingertip taken at different time points after onset of the disease were tested with POC tests. The specificity and sensitivity of the rapid test kits compared to test of reference (RT-PCR) were calculated.

### Results

Among 381 patients with symptoms of COVID-19 who went to the hospital for a diagnostic, 143 patients were RT-PCR negative. Results of test with POC tests were all negative for these patients, indicating a specificity of 100% for both POC tests.

In the RT-PCR positive subgroup (n = 238), 133 patients were tested with COVID-PRESTO® and 129 patients were tested with COVID-DUO® (24 patients tested with both). The further the onset of symptoms was from the date of collection, the greater the sensitivity. The sensitivity of COVID-PRESTO® test ranged from 10.00% for patients having

**Funding:** The study was funded by CHR Orleans (Orléans Regional Hospital Centre), a public hospital with no-profit status, of which all authors are employees. The point-of-care tests were provided free of charge by AAZ-LMB.

**Competing interests:** The point-of-care tests were provided free of charge by AAZ-LMB. This does not alter our adherence to PLOS ONE policies on sharing data and materials.

experienced their 1st symptoms from 0 to 5 days ago to 100% in patients where symptoms had occurred more than 15 days before the date of tests. For COVID-DUO® test, the sensitivity ranged from 35.71% [0–5 days] to 100% (> 15 days).

## Conclusion

COVID-PRESTO® and DUO® POC tests turned out to be very specific (none false positive) and to be sensitive enough after 15 days from onset of symptom. These easy to use IgG/IgM combined test kits are the first ones allowing a screening with CWB sample, by typing from a finger prick. These rapid tests are particularly interesting for screening in low resource settings.

## Introduction

At the end of 2019, a pneumonia of unknown cause detected in Wuhan, China was first reported to the WHO Country Office in China. On January 9th, 2020, the Chinese health authorities and the World Health Organization (WHO) officially announced the discovery of a novel coronavirus, first named 2019-nCoV, then officially termed SARS-CoV-2 (Severe Acute Respiratory Syndrome CoronaVirus 2). This virus, belonging to the coronavirus family, differs from the viruses SARS-CoV, responsible for the SARS outbreak in 2003, and MERS-CoV, responsible for an ongoing outbreak that began in 2012 in the Middle East.

The SARS-CoV-2 virus causes the infectious respiratory disease called COVID-19 (COronaVIrus Disease 2019). This infection mainly results in pneumonia and upper/lower respiratory tract infection. The symptoms of COVID-19 infection appear after an incubation period of approximately 5.2 days [1]. The most common symptoms at onset of COVID-19 illness are fever, cough, and fatigue, but others include headache, sore throat, and even acute respiratory distress syndrome, leading to respiratory failure.

Since the emergence of COVID-19 in China at the end of last year, the SARS-CoV-2 virus has caused a large global outbreak and has become a major worldwide public health issue. The WHO has declared this outbreak a global health emergency at the end of January 2020. On April 12th, 2020, the World Health Organization (WHO) announced that the total global deaths from COVID-19 has surpassed 100 000. Globally, by April 28th, 2020, 2,892,688 cases of COVID-19 have been confirmed and 210,193 patients have died. An estimated 1.7 billion people have been ordered to remain at home as governments take extreme measures to protect their populations.

Due to the rapid spread and increasing number of COVID-19 cases caused by this new coronavirus SARS-CoV-2, rapid and accurate detection of virus and/or disease is increasingly vital to control the sources of infection and prevent the progression of the disease.

Besides the main priority, which is finding an efficient treatment, one of the most important research questions targets the diagnosis of COVID-19. Currently, the real-time RT-PCR (real-time reverse transcriptase polymerase chain reaction) assay is the gold-standard method to detect SARS-CoV-2 [2]. This diagnostic test aims at detecting nucleic acid (RNA) from SARS-CoV-2 in upper and lower respiratory specimens such as nasopharyngeal or oropharyngeal swabs or broncho-alveolar lavage.

In response to the growing COVID-19 pandemic, antibody tests have been developed to detect specific antibodies, IgG and IgM, to SARS-CoV-2 virus in human whole blood, serum or plasma. Two kinds of antibody tests are currently available [3]: quantitative laboratory tests

with antibodies titration by enzyme-linked immunosorbent assay (ELISA) and easy-to-use point-of-care (POC) tests, mainly based on lateral flow chromatographic immunoassays.

COVID-PRESTO® and COVID-DUO® are two POC tests products with CE marking which are marketed by AAZ-LMB. In line with the recommendations of the health authorities, we conducted a prospective observational study to evaluate the performance of both AAZ COVID 19 IgM/IgG POC tests compared to the gold standard, RT-PCR.

## Methods and materials

### Ethics approval

The study was approved by the Orleans Regional Hospital Ethics and Research Committee on March 17th, 2020, and informed consent was obtained from each participant.

### Study population

The study population consisted of adult patients visiting the infectious disease department (Centre Hospitalier Regional Orléans, France) from March, 18th, 2020 to April 10th, 2020. This department receives patients whose symptoms, such as headache, fatigue, fever or respiratory signs suggest a COVID infection, and for whom a diagnosis is requested. Date of onset of symptoms as declared by the patient and age were collected at inclusion. According to severity of disease, patients RT-PCR positive were either hospitalized in the infectious diseases ward, only devoted to treat COVID-19 infected patients, or invited to have regular medical visits in the outpatient consultation. Capillary whole blood (CWB) samples from the fingertip were taken at various stages of the follow-up, even after clinical cure, in order to collect samples from convalescent patients.

### Specimen collection

Nasopharyngeal (NP) swab specimens were collected from patients by trained surveillance officers. A polyester-tipped flexible aluminum-shafted applicator (Microtest M4RT, Remel) was inserted into one of the nostrils until resistance was felt at the nasopharynx, then rotated 180 degrees and withdrawn. After swabbing, the swab applicator was cut off, and each absorbent swab was placed into a vial containing 3 mL of viral transport media. Vials were immediately shipped via a triple packaging system to the virology unit located in the same building of the hospital, then stored if necessary at 4°C for up to 24 hours until testing.

For CWB samples taken at the fingertip, a lancet was used to prick the side of the fingertip to let a large drop of suspended blood form. This blood sample was collected with a 10 µl capillary micropipette that filled automatically. The sample was then expelled by squeezing the micropipette bulb to deposit the blood on the appropriate well of the test cassette. Retesting was performed in a same patient only if the previous test was negative.

### Real-time RT-PCR assays for the detection of SARS-CoV-2

RT-PCR testing of SARS-CoV-2 was performed in Unit of Virology, CHR Orléans. Nucleic acid extraction was performed with automated EZ1 (Qiagen). Specific real-time RT-PCR assays targeting two RNA-dependent RNA polymerases (IP2 and IP4) and E genes were used to detect the presence of SARS-CoV-2 following the instructions in the protocols of the Institut Pasteur and Corman et al., respectively [4, 5]. Amplification was performed on an ABI 7900 Sequence Detection System (Applied Biosystem).

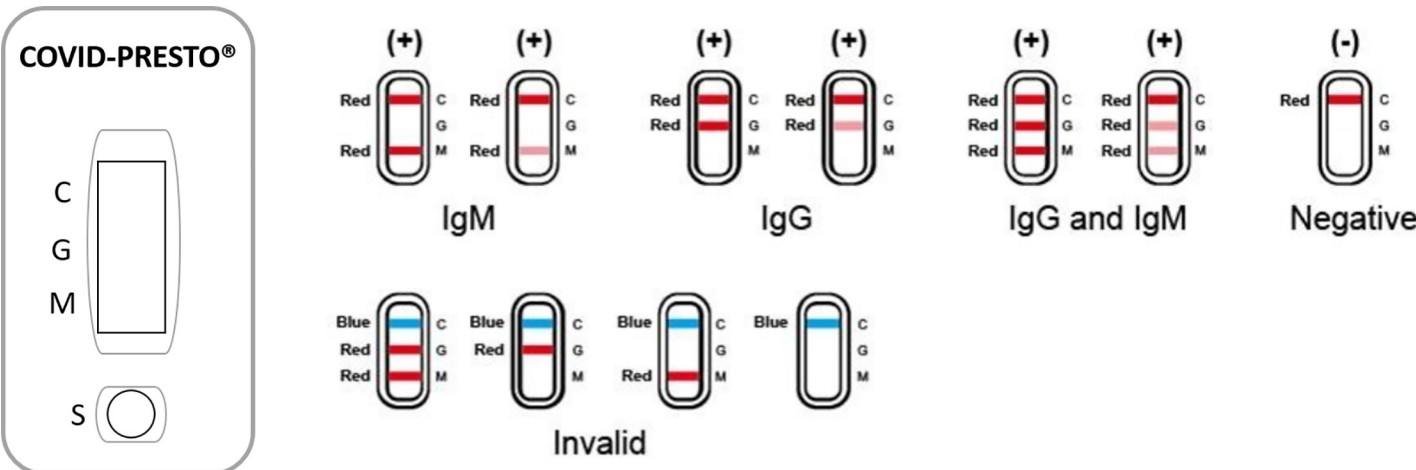

**Fig 1. Interpretation of results for COVID-PRESTO®.**

### Point-of-care tests to be assessed

The SARS-CoV-2 IgG/IgM antibody test kits, COVID-PRESTO® and COVID-DUO®, are targeting on the antibodies specific to N-protein of SARS-CoV-2. They are manufactured and marketed by AAZ-LMB.

Tests were conducted at the site by clinical staff, physicians or nurses, according to manufacturers' instructions. Health workers involved in the study received a two-hours training session for each type of test prior to the beginning of the study.

Both COVID-PRESTO® and COVID-DUO® are lateral flow immune-chromatographic assays (Figs 1 and 2). These tests use anti-human IgM antibody (test line IgM), anti-human IgG antibody (test line IgG) and rabbit IgG (control line C) immobilized on a nitrocellulose strip. The Conjugate (recombinant COVID-19 antigens labeled with colloidal gold) is also integrated into the strip. When a specimen is added to the sample well, followed by assay buffer, IgM and IgG antibodies, if present, will bind to COVID-19 conjugates forming an antigen-antibodies complex.

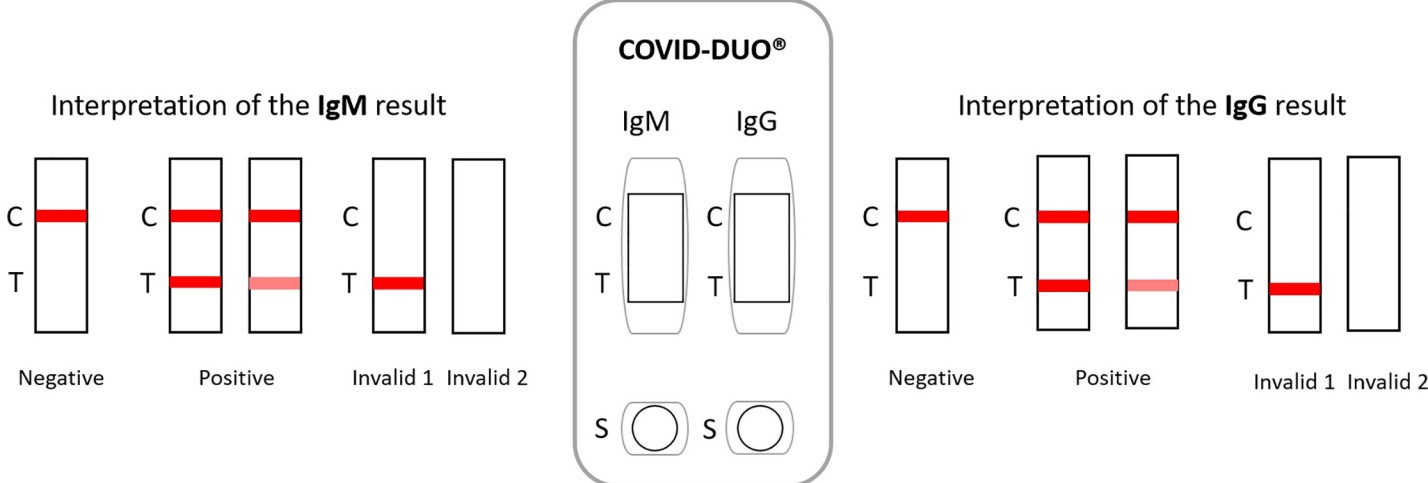

**Fig 2. Interpretation of results for COVID-DUO®.**

This complex migrates through nitrocellulose membrane by capillary action. When the complex meets the line of the corresponding immobilized antibody (anti-human IgM and/or anti-human IgG), the complex is trapped, forming a burgundy colored band which confirms a reactive test result. The result has to be read within 10 minutes by two independent operators. When the control line is the only to be burgundy, the sample is negative. If the control line does not appear, the test is invalid and should be repeated with a new cassette.

## Data analysis

Population were described in terms of %, mean, standard deviation, range and median values.

The test data was analyzed in the Department of Infectiology. The specificity and sensitivity of the rapid test kits compared to test of reference (RT-PCR) were calculated according to the following formulas:

Specificity (%) = 100 x [Negative / (Negative + Positive)].

Sensitivity (%) = 100 x [Positive/ (Positive + Negative)]

Confidence intervals for sensitivity were produced with the Wilson score method [6].

## Results

Overall, 381 patients with symptoms of COVID-19 who went to the hospital for a diagnostic, were included in the study.

RT-PCR was performed in all patients: 62.47% were positive (n = 238). Based on these results, two sub-groups were defined: 143 patients with negative and 238 patients with positive RT-PCR results (Fig 3).

In the negative RT-PCR subgroup, the mean age was 48.20 years (SD: 17.00; range 19–72), median at 46 years. Among these patients, 72 and 71, respectively, were tested with COVID-PRESTO® and COVID-DUO® tests between 24 hours to 8 days from onset of symptoms (median 2 days; range 1–8 days). All results were negative indicating a specificity of 100% for both POC tests.

In the RT-PCR positive subgroup, the mean age of patients was 53.68 years ± 20.18 (median 54; range 19–96).

For COVID-PRESTO® test, CWB samples from the fingertip were collected from 133 patients, only once (n = 133) or at two (n = 16) or three different times (n = 1). Overall 150 samples used to evaluate the sensitivity of this test. The further the onset of symptoms was from the date of collection, the greater the sensitivity (Table 1): 69.23% [CI95%: 53.58–81.43%] for patients with symptoms that occurred from 11 to 15 days before the date of test and 100% [CI95%: 92.59–100%] in patients who experienced first symptoms more than 15 days before the test. Interestingly, among patient with samples collected at two different times, an elderly woman, 75 years of age, with multiple cancer treated by chemotherapy was negative at Day 15 and positive at Day 19, both for IgM and IgG.

For COVID-DUO® test, 129 patients were screened with one (n = 129), two (n = 4) or three samples (n = 1) at different times. The sensitivity was assessed based on 134 conducted tests (Table 2). The sensitivity ranged from 35.71% [CI95%: 16.34–61.24%] for patients having experienced their 1st symptoms from 0 to 5 days ago, to 100% [CI95%: 89.85–100%] in patients where symptoms had occurred more than 15 days before the date of tests.

When considering the distribution of IgM+ and IgG+ patterns among patients with a positive POC test, the IgM were systematically present in the few positive patients with an onset of symptoms from 0 to 5 days ago (n = 2 in COVID-PRESTO® population; n = 5 in COVID-DUO®). The IgM stayed prevalent until 15 days after viral infection while IgG increased over time and became more prevalent after 15 days (Figs 4 and 5).

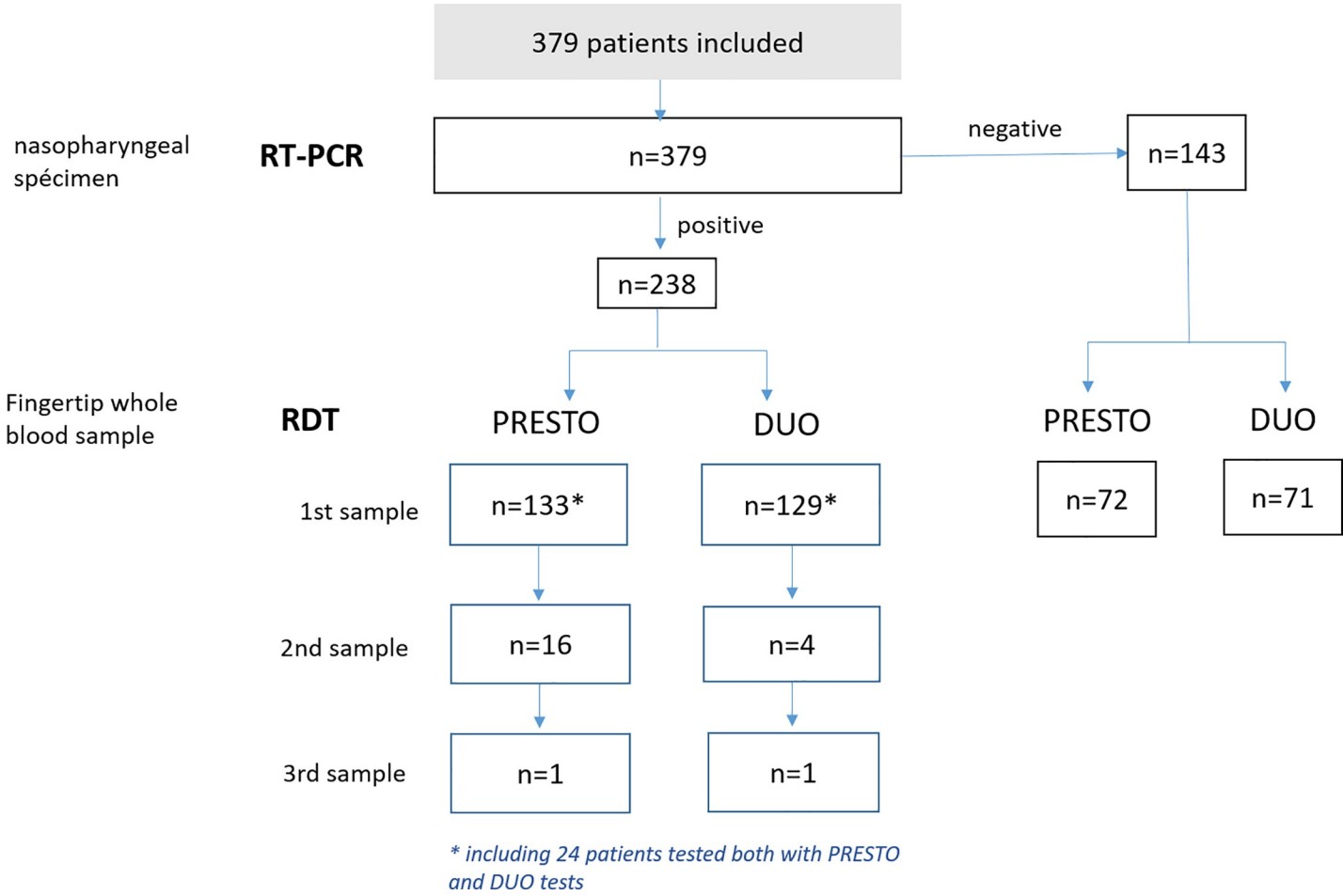

**Fig 3. Number of samples screened with RT-PCR and Point-of-care (POC) tests.**

## Discussion

This prospective observational study aimed at evaluating the performance of two POC tests designed to detect SARS-CoV-2 antibodies IgG and IgM from a CWB sample from the fingertip. We investigated the quick detection approach of COVID-PRESTO® and COVID-DUO® in comparison with RT-PCR testing.

The performance analysis was conducted in 381 patients. The results showed that the sensitivity of both POC tests increases with the duration from symptoms onset, reaching 100% in patients experiencing first symptoms of COVID-19 more than 15 days ago. The specificity of both POC tests was found to be 100%, no false positive results having been obtained.

**Table 1. Evaluation of the sensitivity of the COVID-PRESTO® test.**

|  | Number of days since the onset of symptoms | | | |
|---|---|---|---|---|
|  | **0–5 days** | **6–10 days** | **11–15 days** | **>15 days** |
| Positive | 2 | 25 | 27 | 48 |
| Negative | 18 | 18 | 12 | 0 |
| **Sensitivity** | **10.00%** | **58.14%** | **69.23%** | **100%** |
| [CI 95%] | [2.79–30.10%] | [4.33–71.62%] | [53.58–81.43%] | [92.59–100%] |

**Table 2. Evaluation of the sensitivity of the COVID-DUO® test.**

| | Number of days since the onset of symptoms | | | |
|---|---|---|---|---|
| | **0–5 days** | **6–10 days** | **11–15 days** | **>15 days** |
| Positive | 5 | 23 | 36 | 34 |
| Negative | 9 | 19 | 8 | 0 |
| **Sensitivity** | **35.71%** | **54.76%** | **81.82%** | **100%** |
| [CI 95%] | [16.34–61.24%] | [39.95–68.78%] | [68.04–90.49%] | [89.85–100%] |

The sensitivity and specificity of such strip assays based on immuno-chromatography have been recently estimated in several studies performed with venous blood samples. In a retrospective study, serum from 179 patients was used to detect SARS-CoV-2 IgG/IgM antibodies [7]. Patients were stratified by the time from symptoms onset to sample collection: 0–7 days, 8–15 days and >15 days. Sensitivities of 18.8%, 100% and 100% were reported, respectively, for the three groups with very few patients (n = 8) in the 8–15 days group. The specificity was 77.8%, 50% and 64.3%, respectively, with numerous reported cases of "false positives". In a second prospective study, the sensitivity of a strip assay investigated in 86 patients was 11.1%, 92.9% and 96.8% at the early stage (1–7 days after onset), intermediate stage (8–14 days after onset), and late stage (more than 15 days), respectively [8]. In another prospective study with 397 PCR confirmed COVID-19 patients and 128 negative patients, the performance of another lateral flow immunoassay test product was evaluated [9]. Overall, the sensitivity was 88.66% and the specificity 90.63%. Although this study was performed with more patients (n = 525) than in our study, the evaluation of performance was limited because no information was collected about the period over which each patient had experienced symptoms at the time of blood sample collection. Furthermore, to date, no performance study has been reported based on capillary blood samples.

Although COVID-PRESTO® and COVID-DUO® are only qualitative tests, the reported sensitivities and specificities are closed to those of quantitative assays such as enzyme linked

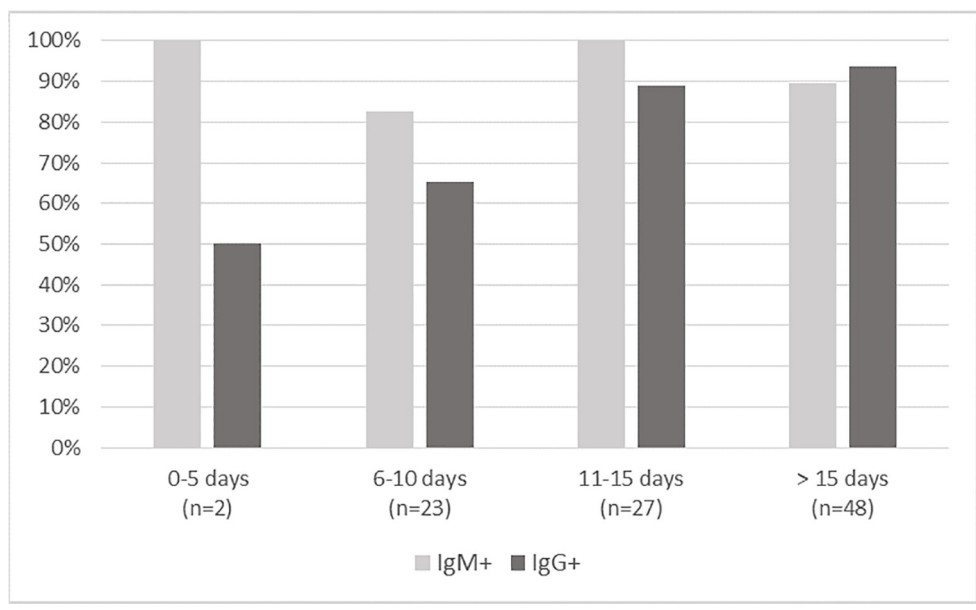

**Fig 4. Patients with a positive COVID-PRESTO® test: Distribution of IgM+ and IgG+ patterns.**

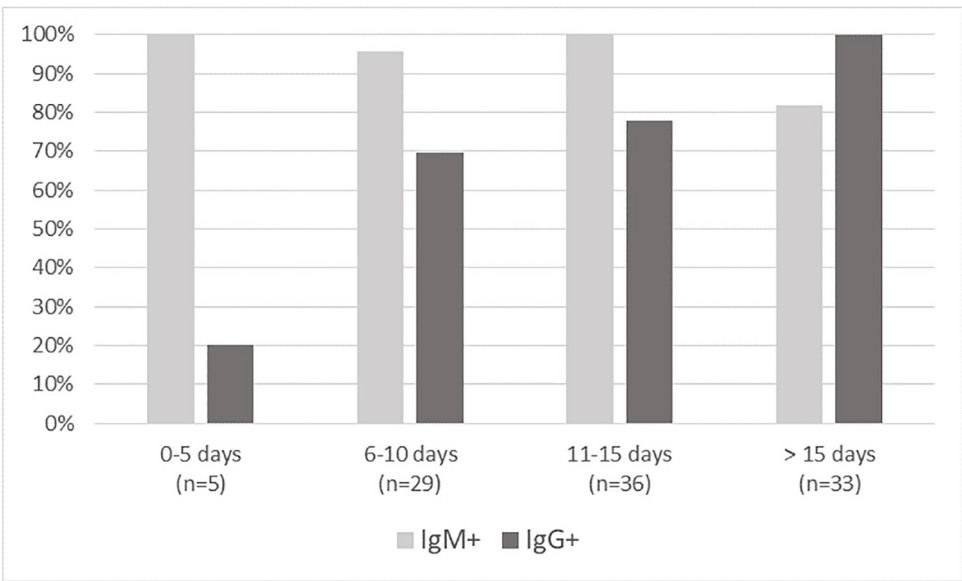

**Fig 5. Patients with a positive COVID-DUO® test: Distribution of IgM+ and IgG+ patterns.**

immunosorbent assay (ELISA). Zhao et al. collected blood samples from 173 patients with a confirmed infection with SARS-CoV-2 (acute respiratory infection syndromes and/or abnormalities in chest CT images accompanied by detectable SARS-CoV-2 RNA) at different times after onset of COVID-19: <7 days since onset (early phase), 8–14 days after onset (middle phase) and 13–39 days after onset (later phase) [10]. The detection of IgM and IgG against SARS-CoV-2 in this study was performed using ELISA kits. The sensitivities of IgM assays were 38.3%, 73.3% and 94.3% successively, among samples from patients in early, middle and later phases, respectively. For IgG, the values were 38.3%, 54.1% and 79.8%. Interestingly, the RNA test (RT-PCR on samples from the respiratory tract) had the highest sensitivity (66.7%) in the early phase of illness while RNA was only detectable in 45.5% of samples of day 15–39. From a methodological point of view, the performance study presented here was more robust to that of Zhao et al. because the positive population used as reference to evaluate the sensitivity of POC tests was only based on positive RT-PCR results, and not a mix between syndromes, imagining findings and RNA detection.

The results of the present study highlight two major points. First, similar to ELISA tests the sensitivity of POC tests increases when the sample is collected further from the symptom onset. Second, these tests (either qualitative or quantitative) can help to diagnose a past infection after elimination of the virus by the immune system. Therefore, rapid POC testing with CWB samples can provide similar epidemiological information as immunoassay tests [11], but with a lower cost and easier implementation, thus facilitating a larger coverage.

Currently, the extent and the time kinetics of humoral response against SARS-CoV-2 are not known. It is widely accepted that IgM is usually the first responded antibody providing the first line of defense during viral infections, prior to the generation of adaptive, high affinity IgG responses serving as the more robust long term immunity. We were not able to study the humoral response at the individual level because too few patients could have been tested more than once. At the population level, the patterns of IgM/IgG results obtained for positive tests with COVID-DUO® made it possible to perceive a constant dominant presence of IgM which was surpassed by the progressive appearance of IgG from 15 days after symptoms onset. This coincided with our observations with the COVID-PRESTO®. One of the reasons could lie on

the high proportion (90%) of false negative results during the early phase of infection, directly linked to the low titers of antibodies during the first days after infection. Both IgM and IgG titers were found to be low or undetectable 4 days after infection [12, 13]. It was also shown that the presence of antibodies was less than 40% among patients within 1 week since onset, and rapidly increased to 94.3% (IgM) and 79.8% (IgG) from day-15 after onset [10]. The presence of IgM and IgG antibodies against SARS-CoV-2 within 2 weeks from the onset of symptoms was confirmed by others [12, 14]. Recently, in 41 COVID-19 patients confirmed by RT-PCR, it was shown by chemiluminescent immunoassay that the median time of seroconversion was 11 days after disease onset for IgG and 14 days for IgM [15]. The time required to have detectable levels of antibodies explains the poor performance (sensitivity 18.4%) reported for antibody tests evaluated in acute patients enrolled from the emergency room, of which only 7 of 38 RT-PCR-positive samples gave positive results for a COVID-19 IgM/IgG Rapid Test [16]. From this study, Cassaniti *et al.* concluded that the Rapid Test lateral Flow Immuno Assay was not recommended for triage of patients with suspected COVID-19 as the disease cannot be excluded when viral serological testing is negative. Although slightly lower than the specificity obtained for COVID-PRESTO® and–DUO®, the specificity demonstrated by Cassaniti *et al.* at early stages was also high (91.7%), with only one false negative result among 12 tests on RT-PCR negative samples.

This study has several limitations. First, the date of onset of symptoms related to SARS-CoV-2 infection implied recall of facts from memory. This recall bias could lead to some imprecise classification when stratifying the samples by days between onset of symptoms and date of blood samples. Second, few patients with a negative serology could have been re-tested with a second blood sample. In these conditions, we were not able to study the dynamics of seroconversion on individual level. Third, there were still negative tests in RT-PCR positive patients up to 15 days after onset. The reasons are multiple and include the relatively low titers of antibody in the early stages of infection as reported by others [17] and the difference in individual immune response antibody production. Lastly, the strength of antibody response depends on several factors, including age, severity of disease, and certain conditions like immunodeficiency disorders. Therefore it would have been interesting to stratify the population depending on immune health. Indeed, we had few subjects with profound immunosuppression who were still negative 15 days after onset. We know, however, that seroconversion could occur later in such patients [18, 19]. Future studies should focus on seroconversion from Day 15 to Day 30 in highly immunocompromised patients infected with COVID-19. However, the highly immunosuppressed patient in this study was well documented to seroconvert between day 15 and day 19, which provides reassurance of the performance of the POC tests, even in this population. Furthermore, CWB samples collected from patients with other respiratory infections could also be investigated in future studies, in order to further investigate the specificity of the two POC tests and exclude any cross-reactivity with other virus infections, particularly of the coronavirus family.

Despite these limitations, COVID-PRESTO® and DUO® POC tests turned out to be very specific (none false positive) and to be sensitive enough after 15 days from onset of symptom. These easy to use IgG/IgM combined test kits are the first ones allowing an epidemiological screening using CWB samples from the fingertip in order to determine the seroprevalence in a large asymptomatic population. The tests are simple, qualitative, visually interpretable, and give a result within 15 minutes. A positive serology allows to determine whether a person has already been infected by SARS-CoV-2. Serologic tests will be needed to assess the response to vaccine candidates and to map levels of immunity in communities. These rapid tests are particularly interesting for low resource settings such as at the bedside or any other locations where lab tests are less obvious.

## Supporting information

**S1 Raw data.**
(XLSX)

## Acknowledgments

The authors would like to thank the technical staff of the Department of Infectious diseases for excellent assistance. Furthermore, the authors thank Angèle Guilbot of Clinact, France for providing medical writing support/editorial support in accordance with Good Publication Practice (GPP3) guidelines.

## Author Contributions

**Conceptualization:** Thierry Prazuck, Jérôme Guinard, Gilles Pialoux, Laurent Hocqueloux.

**Data curation:** Thierry Prazuck, Jérôme Guinard, Gilles Pialoux, Laurent Hocqueloux.

**Formal analysis:** Thierry Prazuck, Mathilda Colin, Jérôme Guinard, Laurent Hocqueloux.

**Investigation:** Thierry Prazuck, Mathilda Colin, Susanna Giachè, Camélia Gubavu, Aymeric Seve, Vincent Rzepecki, Marie Chevereau-Choquet, Catherine Kiani, Victor Rodot, Elsa Lionnet, Laura Courtellemont, Jérôme Guinard, Laurent Hocqueloux.

**Methodology:** Thierry Prazuck, Jérôme Guinard, Gilles Pialoux, Laurent Hocqueloux.

**Project administration:** Thierry Prazuck.

**Resources:** Thierry Prazuck, Mathilda Colin, Camélia Gubavu, Laurent Hocqueloux.

**Supervision:** Thierry Prazuck, Mathilda Colin, Camélia Gubavu, Laurent Hocqueloux.

**Writing – original draft:** Thierry Prazuck, Jérôme Guinard, Gilles Pialoux, Laurent Hocqueloux.

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
