## [Decision Letter · Decision Letter 0]

1 Jul 2020

PONE-D-20-15374

Evaluation of performance of two SARS-CoV-2 Rapid whole-blood finger-stick IgM-IgG Combined Antibody Tests

PLOS ONE

Dear Dr. Guilbot,

Thank you for submitting your manuscript to PLOS ONE. After careful consideration, we feel that it has merit but does not fully meet PLOS ONE’s publication criteria as it currently stands. Therefore, we invite you to submit a revised version of the manuscript that addresses the points raised during the review process.

We look forward to receiving your revised manuscript.

Kind regards,

Kwok Hung Chan, Ph.D

Academic Editor

PLOS ONE

Journal Requirements:

2.Thank you for including your ethics statement:   "The study was approved by the local Ethics Committee on March 17th 2"020, and informed consent was obtained from each participant."

3.Thank you for stating the following financial disclosure:

 [Rapid Diagnostic Tests were provided free of charge by AAZ-LMB.].

4.We note that you have indicated that data from this study are available upon request. PLOS only allows data to be available upon request if there are legal or ethical restrictions on sharing data publicly. For information on unacceptable data access restrictions, please see http://journals.plos.org/plosone/s/data-availability#loc-unacceptable-data-access-restrictions.

5. Please amend your authorship list in your manuscript file to include author Angèle Guilbot.

Additional Editor Comments (if provided):

The specificity of the POCT (RDTs) which depends on COVID-19 negative patients by RT-PCR is not enough. The specificity test must be included patients with other respiratory virus infection including human coronaviruses.

Reviewers' comments:

Reviewer's Responses to Questions

**Comments to the Author**

1. Is the manuscript technically sound, and do the data support the conclusions?

Reviewer #1: Yes

Reviewer #2: Yes

2. Has the statistical analysis been performed appropriately and rigorously? 

Reviewer #1: Yes

Reviewer #2: Yes

3. Have the authors made all data underlying the findings in their manuscript fully available?

Reviewer #1: Yes

Reviewer #2: Yes

4. Is the manuscript presented in an intelligible fashion and written in standard English?

Reviewer #1: Yes

Reviewer #2: Yes

5. Review Comments to the Author

Reviewer #1: Review

This manuscript compares performance of two Point-of-care rapid whole blood finger stick antibody tests for COVID-19. Authors were able to show the performance of these antibody tests (sensitivity and specificity) is good when used after 15 days from onset of symptom

Major Concerns:

1. I have concerns with authors calling COVID-19 antibody tests as “Rapid Diagnostic Tests” (RDTs). WHO, USA CDC, and USA FDA have clearly discouraged using antibody tests as diagnostic tests because of test performance issues and low prevalence of disease. The appropriate way to address these lateral flow antibody tests would be to call them “Point-of-care (POC) tests”.

2. In my opinion, this study is not a “real-life study”, the authors are not studying the effectiveness of intervention (lab results) in “real-life” in the enrolled patients. Authors do not describe how the results of antibody tests were used for providing patient care in this manuscript to qualify as “real-life” study. This study is better characterized as a “Prospective Observational Study”.

Minor Concerns

1. Cover page lists “Angèle Guilbot, Ph.D” as corresponding author and page 1 lists “Thierry Prazuck” as corresponding author. Please verify corresponding author.

2. Please use abbreviations consistently. First use of abbreviation should be spelled out in both abstract and main body separately. Eg. SARS-CoV-2 never spelled out as “Severe acute respiratory syndrome coronavirus 2”. COVID-19 should be spelled out as “coronavirus disease 2019”. RT-PCR should be spelled out as “real-time reverse transcriptase polymerase chain reaction” during first use.

3. Page 3, Line # 92-95:

a. Consider using “antibody tests” instead of “serologic tests” because serologic tests also include antigen tests.

b. Two types of antibody tests are “Point-of-care (POC) tests” (generally lateral flow devices) and “Laboratory tests” (ELISA or chemiluminescent immunoassay tests). Please refer to CDC website for antibody testing classification, https://www.cdc.gov/coronavirus/2019-ncov/lab/resources/antibody-tests-guidelines.html.

c. Consider changing “Rapid Diagnostic Tests (RDTs)” to “Point-of-care (POC) tests” (throughout the manuscript).

4. Page 4, line # 113, consider changing “Fingertip blood samples” to “capillary whole blood (WB) sample from fingertip” (throughout the manuscript).

5. Page 4, Line # 136: Consider changing “Rapid Diagnostic Tests” to “Point-of-care (POC) tests” (throughout the manuscript).

6. Page 6, line # 172-175: Consider providing details of average day from symptoms onset (at least average and range).

7. Page 6, line # 192-196: Not sure if authors can draw the conclusion “the IgM were the first antibodies to be detected” from the data provided in Figure 4 and 5. IgG antibodies were present at 0-5 days along with IgM for both COVID-PRESTO® and COVID-DUO®. The % was lower but IgG were not absent at 0-5 days. Also, the number of specimens tested was too small to come to such conclusions.

8. Page 8, line # 239-245:

a. Consider rephrasing the statement “Firstly, as for the assessed RDT, the sensitivity of ELISA tests increases with the duration from symptom onset. Moreover, we showed that direct antibody typing with whole blood is as sensitive as immunoassay performed with serum in a retrospective way.” My understanding is that the authors are trying to say, similar to ELISA tests, the sensitivity of Point of care (lateral flow) antibody tests increases when specimen is collected further from the symptom onset.

b. I disagree with the second point that authors are making “Secondly, these diagnostic tests (either qualitative or quantitative) can help to diagnose a past infection after elimination of the virus by the immune system.” This study does not show diagnostic application of antibody tests. This generalized statement is premature without defining boundaries and criteria for using antibody tests to “diagnose past infections” of COVID-19. Clinical utility of diagnosing past COVID-19 infections is very limited (eg. negative molecular tests in patients presenting late in illness and to establish diagnosis of late complications of COVID-19 illness like Multisystem Inflammatory Syndrome in children) and should be clearly defined before calling antibody tests as “diagnostic tests”.

c. Authors do try to explain use of antibody tests by saying “Thus, combining RT-PCR and antibody detection allows to largely diagnose COVID-19 people regardless of the delay between infection and diagnosis”, but this is till too vague description of its use. This statement is true if antibody tests are used in symptomatic patients only.

9. Page 8, line # 251-253: The results of the study do not show “the switch between the first production of IgM and the later onset of IgG.” As shown in Figure 4 and 5, both IgG and IgM were present throughout the duration from onset of symptoms (0-5 days to >15 days). There was no data provided which showed “seroconversion” or “switch” in patients tested.

10. Page 8, line # 267-269: Consider rephrasing statement “Although slightly lower than the specificity obtained for COVID-PRESTO® and –DUO®, the specificity at early stages was high (91.7%), preventing false positive diagnoses.”

a. Not sure what authors are trying to convey here. Because of high specificity, antibody tests will have low “false positive” diagnosis but more importantly these antibody tests will have significantly high numbers of “false negatives” because of low sensitivity in early stages of disease.

b. Please provide more details of calculations to understand how authors came up with specific of 91.7%. Please also define what does “early stage” means (0-5 days or 0-10 days)?

11. Page 9, line 289-291: Authors say that the antibody tests can be used for “screening” but do not explain screening for what purpose? Screening for diagnosis or immunity or epidemiologic studies, etc.?

12. Page 9, line 292-296: the conclusion is completely different than rest of the manuscript. Throughout the manuscript authors are presenting antibody tests as “diagnostic tests” and then at the end they appropriately limit the use to “assess the response to vaccine candidates and to map levels of immunity in communities”. These two uses are Epidemiologic use of antibody tests and are not considered as “diagnostic use”.

Reviewer #2: This is well written manuscript.

Please change verb responsible (line 68) to causing or similar to omit repeating (previous sentence already has responsible)

My preference would be to use first, second, third etc.(lines 239, 242, 270, 273, 275).

Figure 3 can be omitted

6. PLOS authors have the option to publish the peer review history of their article (what does this mean?). If published, this will include your full peer review and any attached files.

Reviewer #1: **Yes: **Gagan Mathur, MD, MBA

Reviewer #2: No

---

## [Author Response · Author response to Decision Letter 0]

17 Jul 2020

We would like to thank the editor and the reviewers for their thorough and constructive review of our manuscript. We considered all of your requests very carefully and provide our responses in blue below as well as, if applicable, directly in the manuscript.

Answers to Editor’s comments:

1. PLOS ONE style requirements: The style requirement have been verified.

2. Updated ethics statement (also corrected in the manuscript): “The study was approved by the Orleans Regional Hospital Ethics and Research Committee on March 17th 2020, and informed consent was obtained from each participant.”

3. Financial disclosure statement: “The study was funded by CHR Orleans (Orléans Regional Hospital Centre), a public hospital with no-profit status, of which all authors are employees. The point-of-care were provided free of charge by AAZ-LMB.” 

4. Data availability: The data is available via APREMIT45, Direction de la recherche, Centre Hospitalier Regional Orleans, 14 avenue de l’hôpital 45067 Orleans

5. List of authors/role of Angèle Guilbot: As the submission of this manuscript has been delegated to Clinact. Therefore Angèle Guilbot will act as primary contact person regarding the manuscript processing in the journal, i.e. serve as corresponding “author” for the submission procedure only. She will not appear as corresponding author on the manuscript.

6. The specificity test must be included patients with other respiratory virus infection including human coronaviruses: This is indeed relevant when evaluating performances on serum. In real life, when testing patients’ blood samples from the fingertip, it is actually not possible to find any patient with current active corononavirus infection different from COVID19.

Answers to Reviewer#1 comments:

1. Concerns regarding “Rapid Diagnostic Tests” (RDTs): We agree and have renamed these “Point-of-care (POC) tests” throughout the manuscript according to your suggestion. 

2. Concerns regarding “real-life study”: “Prospective Observational Study”: We agree and have renamed this “Prospective Observational Study” throughout the manuscript according to your suggestion.

3. List of authors/role of Angèle Guilbot: As the submission of this manuscript has been delegated to Clinact. Therefore Angèle Guilbot will act as primary contact person regarding the manuscript processing in the journal, i.e. serve as corresponding “author” for the submission procedure only. She will not appear as corresponding author on the manuscript.

4. Consistency of abbreviations: We verified all abbreviations to make sure they are spelled out the first time used.

5. Page 3, Line # 92-95 : 

a. Concerns regarding “serologic tests” Renamed to “antibody tests”.

b. Definition of two types of antibody tests: Rephrased.

c. Concerns regarding “Rapid Diagnostic Tests (RDTs)”: Renamed.

6. Page 4, line # 113, Concerns regarding “Fingertip blood samples” Renamed.

7. Page 4, Line # 136: Concerns regarding “Rapid Diagnostic Tests” Renamed.

8. Page 6, line # 172-175: Details of average day from symptoms onset Added median (2 days) and range (1-8 days).

9. Page 6, line # 192-196: Concerns regarding conclusion “the IgM were the first antibodies to be detected”: We agree, the conclusion was rephrased.

10. Page 8, line # 239-245:

a. Concerns regarding statement of the sensitivity: We agree, the sentence was rephrased.

b. Concerns regarding statement of diagnostic application We agree, it was deleted.

c. Authors do try to explain use of antibody tests by saying “Thus, combining RT-PCR and antibody detection allows to largely diagnose COVID-19 people regardless of the delay between infection and diagnosis”, but this is till too vague description of its use. This statement is true if antibody tests are used in symptomatic patients only. We agree, the paragraph was rephrased.

11. Page 8, line # 251-253: Concerns regarding description of IgG and IgM detection: We agree, the sentence was rephrased.

12. Page 8, line # 267-269: Concerns regarding statement of specificity: We agree, it was rephrased.

a. Not sure what authors are trying to convey here. Because of high specificity, antibody tests will have low “false positive” diagnosis but more importantly these antibody tests will have significantly high numbers of “false negatives” because of low sensitivity in early stages of disease. We agree, the paragraph was rephrased.

b. Please provide more details of calculations to understand how authors came up with specific of 91.7%. Please also define what does “early stage” means (0-5 days or 0-10 days)? More details were included.

13. Page 9, line 289-291: Request for more information regarding “screening”: The claim has been specified.

14. Page 9, line 292-296: the conclusion is completely different than rest of the manuscript. Throughout the manuscript authors are presenting antibody tests as “diagnostic tests” and then at the end they appropriately limit the use to “assess the response to vaccine candidates and to map levels of immunity in communities”. These two uses are Epidemiologic use of antibody tests and are not considered as “diagnostic use”. We agree and addressed the “diagnostic application” throughout the rest of the manuscript.

Answers to Reviewer#2 comments:

1. Please change verb responsible (line 68) to causing or similar to omit repeating (previous sentence already has responsible): Done.

2. My preference would be to use first, second, third etc.(lines 239, 242, 270, 273, 275): Agreed and changed.

3. Figure 3 can be omitted: In this case we disagree. In our opinion the figure summarizes and visualizes the design of the study and we would like to keep it. As several journals request this type of study flow-chart, we leave the final decision of the editor.

---

## [Decision Letter · Decision Letter 1]

27 Jul 2020

PONE-D-20-15374R1

Evaluation of performance of two SARS-CoV-2 Rapid IgM-IgG Combined Antibody Tests on capillary whole blood samples from the fingertip

PLOS ONE

Dear Dr. Guilbot,

Thank you for submitting your manuscript to PLOS ONE. After careful consideration, we feel that it has merit but does not fully meet PLOS ONE’s publication criteria as it currently stands. Therefore, we invite you to submit a revised version of the manuscript that addresses the points raised during the review process.

We look forward to receiving your revised manuscript.

Kind regards,

Kwok Hung Chan, Ph.D

Academic Editor

PLOS ONE

Additional Editor Comments (if provided):

Serum samples collected from other respiratory infections can be used to replace blood samples to show the specificity of these two antibody rapid kits. So, the lack of fingertip blood samples from current patients with other respiratory virus infections is a major limitation of this manuscript and should be included in the discussion. This indicates cross-reactivity with other virus infections particularly coronavirus family can’t be excluded.

Reviewers' comments:

Reviewer's Responses to Questions

**Comments to the Author**

1. If the authors have adequately addressed your comments raised in a previous round of review and you feel that this manuscript is now acceptable for publication, you may indicate that here to bypass the “Comments to the Author” section, enter your conflict of interest statement in the “Confidential to Editor” section, and submit your "Accept" recommendation.

Reviewer #1: All comments have been addressed

Reviewer #2: All comments have been addressed

2. Is the manuscript technically sound, and do the data support the conclusions?

Reviewer #1: Yes

Reviewer #2: Yes

3. Has the statistical analysis been performed appropriately and rigorously? 

Reviewer #1: Yes

Reviewer #2: Yes

4. Have the authors made all data underlying the findings in their manuscript fully available?

Reviewer #1: Yes

Reviewer #2: Yes

5. Is the manuscript presented in an intelligible fashion and written in standard English?

Reviewer #1: Yes

Reviewer #2: Yes

6. Review Comments to the Author

Reviewer #1: All concerns satisfactorily addressed by the authors in the revision. Ok to proceed with publication

Reviewer #2: Thank you for addressing all comments. I would suggest final proofread by the native English speaker to address minor things throughout manuscript. Please change COronaVirus disease to coronavirus disease (lines 27, 72)

7. PLOS authors have the option to publish the peer review history of their article (what does this mean?). If published, this will include your full peer review and any attached files.

Reviewer #1: **Yes: **Gagan Mathur, MD, MBA

Reviewer #2: No

---

## [Author Response · Author response to Decision Letter 1]

30 Jul 2020

A cover letter responding to the editor's comment and an excel file with the raw data as supporting information have been uploaded.

---

## [Editor Report · Decision Letter 2]

3 Aug 2020

Evaluation of performance of two SARS-CoV-2 Rapid IgM-IgG Combined Antibody Tests on capillary whole blood samples from the fingertip

PONE-D-20-15374R2

Dear Dr. Guilbot,

We’re pleased to inform you that your manuscript has been judged scientifically suitable for publication and will be formally accepted for publication once it meets all outstanding technical requirements.

Kind regards,

Kwok Hung Chan, Ph.D

Academic Editor

PLOS ONE
---

## [Editor Report · Acceptance letter]

8 Sep 2020

PONE-D-20-15374R2 

Evaluation of performance of two SARS-CoV-2 Rapid IgM-IgG Combined Antibody Tests on capillary whole blood samples from the fingertip 

Dear Dr. Prazuck:

I'm pleased to inform you that your manuscript has been deemed suitable for publication in PLOS ONE. Congratulations! Your manuscript is now with our production department. 

Kind regards, 

on behalf of

Dr. Kwok Hung Chan 

Academic Editor

PLOS ONE